# Mechanochemically responsive polymer enables shockwave visualization

Polette J. Centellas [1], Kyle D. Mehringer[2], Andrew L. Bowman[3], Katherine M. Evans [1], Parth Vagholkar[2], Travis L. Thornell [3], Liping Huang[4], Sarah E. Morgan [2], Christopher L. Soles [1], Yoan C. Simon [5] ✉ & Edwin P. Chan [1] ✉

Understanding the physical and chemical response of materials to impulsive deformation is crucial for applications ranging from soft robotic locomotion to space exploration to seismology. However, investigating material properties at extreme strain rates remains challenging due to temporal and spatial resolution limitations. Combining high-strain-rate testing with mechanochemistry encodes the molecular-level deformation within the material itself, thus enabling the direct quantification of the material response. Here, we demonstrate a mechanophore-functionalized block copolymer that self-reports energy dissipation mechanisms, such as bond rupture and acoustic wave dissipation, in response to high-strain-rate impacts. A microprojectile accelerated towards the polymer permanently deforms the material at a shallow depth. At intersonic velocities, the polymer reports significant sub-surface energy absorption due to shockwave attenuation, a mechanism traditionally considered negligible compared to plasticity and not well explored in polymers. The acoustic wave velocity of the material is directly recovered from the mechanochemically-activated subsurface volume recorded in the material, which is validated by simulations, theory, and acoustic measurements. This integration of mechanochemistry with microballistic testing enables characterization of high-strain-rate mechanical properties and elucidates important insights applicable to nanomaterials, particle-reinforced composites, and biocompatible polymers.

The response of a material to an impulsive event is an important scientific problem where different fields hold seemingly opposing views. For example, planetary science[1] and geophysics[2] communities understand high-velocity impacts through the lens of shockwave physics and highlight the importance of wave mechanics (i.e., wave propagation and attenuation) within the impacted material. In contrast, impact engineering[3,4] and materials science[5–11] communities approach the same problem through the lens of impact physics where emphasis is placed on the inelastic failure properties of the material (i.e., plasticity or fracture). Bridging these different perspectives is a challenging endeavor due to inaccessible in-situ volumetric measurements of the impacted site. Analyzes are generally limited to post-mortem surface

[1]Materials Science and Engineering Division, National Institute of Standards and Technology, Gaithersburg, USA. [2]School of Polymer Science and Engineering, University of Southern Mississippi, Hattiesburg, USA. [3]Geotechnical and Structures Laboratory, US Army Engineer Research and Development Center, Vicksburg, USA. [4]Department of Materials Science and Engineering, Rensselaer Polytechnic Institute, Troy, USA. [5]School of Molecular Sciences and Biodesign Center for Sustainable Macromolecular Materials and Manufacturing, Arizona State University, Tempe, USA. ✉e-mail: yoan.simon@asu.edu; edwin.chan@nist.gov

measurements due to poor spatial resolution and inadequate temporal resolution in macroscale and microscale impact studies, respectively[7].

One promising material platform that can reconcile these differing views on how materials respond to impulsive deformation is mechanophores[12–16]. Mechanophores, in this context, are force-sensitive molecules that comprise a mechanically labile or isomerizable chemical bond. Upon rupture of this bond, the activated mechanophore undergoes a chemical transformation such as color change, fluorescence activation, or release of small molecules[17–20]. Once incorporated in a polymer matrix, mechanophores offer the requisite spatial and temporal resolution to visualize the impulsive deformation of a material that cannot be captured otherwise.

In this work, we demonstrate the application of a functional mechanophore block copolymer system in microballistic impact studies to directly visualize and quantify the material response at extreme deformation rates. The integrated mechanophores successfully self-report a level of information that far exceeds traditional post-mortem failure analyzes. We show that the energy dissipated by the mechanophore-functionalized material scales with the microballistic impact velocity and identify a transition from plastic deformation to shockwave dominant behavior. Within the shockwave dominant regime, the high-velocity impact event induces significant subsurface energy absorption due to Mach cone formation. The Mach cone is chemically encoded by our mechanophore system within the material and characterized via fluorescence microscopy. From fluorescence microscopy measurements, we can accurately determine the acoustic wave velocity of the material and validate via finite element simulations, contact mechanics theory, and acoustic measurements. Importantly, our integration of polymer mechanochemistry, polymer self-assembly, and microballistic impact experiments uncovers unexpected insights into how a material responds mechanically to a shock event, information that cannot be experimentally captured via other methodologies.

## Results and discussion

As a model system, we utilize a maleimide-anthracene (MA) mechanophore, which is activated upon mechanically-induced cleavage of the MA moiety into two species, as illustrated in Fig. 1a[21–25]. As previously reported, there are various approaches to mechanically induce bond cleavage of mechanophores in polymeric materials, including ultrasonication[23,26,27], quasi-static tensile[22] and compression[21] testing, and high-strain-rate compression[14,15] and shockwave loading[12] experiments. The activated MA mechanophore exhibits a detectable fluorescence signal under UV-A irradiation for at least two days with thermal stability up to 200 °C[21]. To ensure uniform spatial and orientational distribution and thus increase the probability of activation[17,19], the MA mechanophore is synthetically incorporated into the junction point of a polyisobutylene-$b$-polystyrene (PIB-$b$-PS) diblock copolymer (Fig. 1b). This synthetic strategy enables the maleimide-anthracene-functionalized block copolymer (MA-BCP) to self-assemble into phase-separated domains with the mechanophores positioned at the interfaces between the PS and PIB phases (Fig. 1c). Based on atomic force microscopy (AFM) phase imaging and small angle X-ray scattering (SAXS) data (Supplementary Fig. 1), we conclude that this block copolymer self-assembles into a spherical morphology consisting of isolated PS spheres in a continuous PIB matrix.

To induce a shock event, we use laser-induced projectile impact testing (LIPIT)[5–10] to launch individual silica microprojectiles (20 $\mu m$ diameter) towards relatively thick ($\approx 50\,\mu m$) MA-BCP film surfaces with impact velocities ($v_i$) ranging from approximately 100 ms$^{-1}$ to 520 ms$^{-1}$. An ultrafast camera captures the trajectory of the microprojectile as it approaches and then rebounds away from the MA-BCP film (Fig. 2a and Supplementary Fig. 2a). The contact duration between the microprojectile and film is < 20 ns (Supplementary Fig. 2b) and the penetration strain rate, approximated as the ratio of impact velocity to contact depth[5,28], is on the order of 10$^8$ s$^{-1}$ (Supplementary Table 1). The rebound velocity ($v_{rb}$) is slower than the impact velocity for all tests, indicating inelastic collisions ($v_i > v_{rb}$) and kinetic energy transfer from the microprojectile to the film (Fig. 2b). This energy transfer, which characterizes the energy dissipated by the film, is quantified by the kinetic energy loss parameter ($\Delta KE/KE_i$)[3].

Figure 2c shows two regimes for the kinetic energy loss parameter as a function of impact velocity. At low velocities, the energy dissipated by the film monotonically increases with impact velocity. Around $v_i \approx 300$ ms$^{-1}$, the response transitions into the second regime where the loss parameter no longer shows a strong

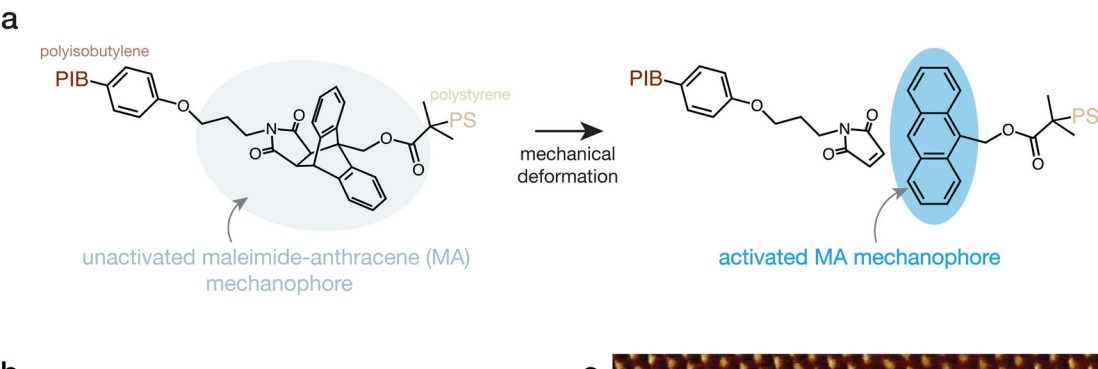

**a**

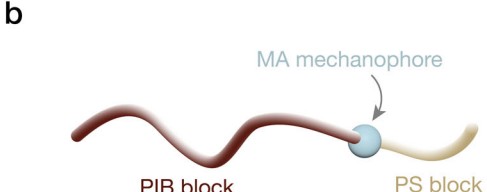

**b**

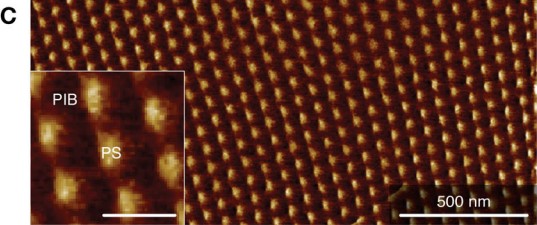

**c**

**Fig. 1 | The chemistry and structure of the maleimide-anthracene-functionalized block copolymer (MA-BCP) material. a** Chemical scheme depicting mechanophore activation upon the MA bond rupture due to mechanical deformation. **b** Schematic of the diblock copolymer with MA mechanophore incorporated between polyisobutylene (PIB) and polystyrene (PS) blocks. **c** AFM phase image of the sample surface showing spherical features attributed to the PS phase distributed within the PIB matrix. Scale bar for the magnified inset image is 100 nm.

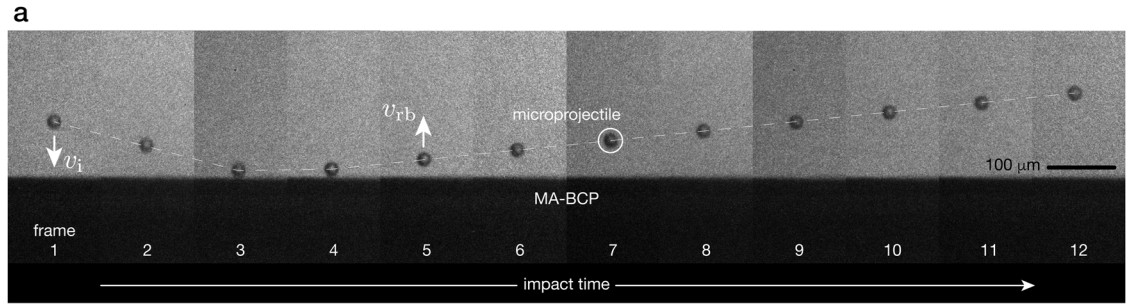

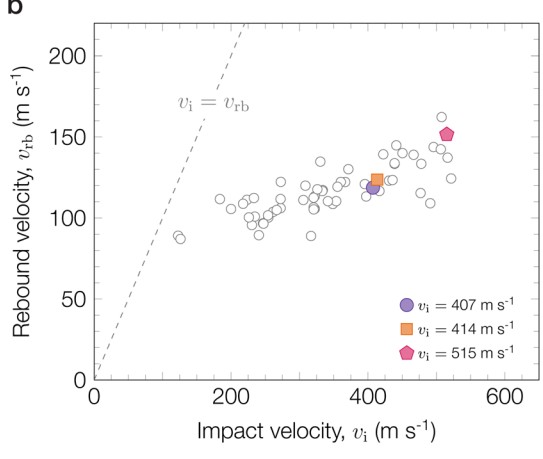

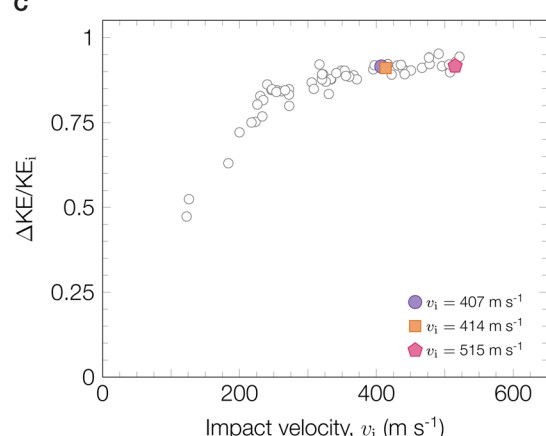

**Fig. 2 | High-strain-rate microballistic impact test results of the MA-BCP films.**
**a** Time-lapse ultrafast camera images of a representative impact event (interframe time is 230 ns). Images capture the trajectory of a 20 $\mu$m silica microprojectile as it impacts and rebounds off the MA-BCP film surface. **b** Plot of impact ($v_i$) and rebound ($v_{rb}$) microprojectile velocities measured from the ultrafast camera images. **c** Plot of the kinetic energy loss parameter ($\Delta KE/KE_i$) as a function of impact velocity. Each open circle corresponds to one impact test. The three filled symbols highlight specific impact tests, $v_i = 407$ ms$^{-1}$ (purple circle), 414 ms$^{-1}$ (orange square), and 515 ms$^{-1}$ (red pentagon), where the results are discussed in greater detail in Fig. 3 and Fig. 4.

dependence with impact velocity and $\Delta KE/KE_i > 0.90$. There is appreciable kinetic energy dissipated by the film for impacts in this second regime. We estimate that the contact duration between the microprojectile and MA-BCP film is much shorter than the time required for one acoustic wave to reflect across the film thickness (Supplementary Fig. 2c), therefore, we do not need to consider the interference of reflected waves. Under this condition, the only two mechanisms available to dissipate energy are acoustic wave attenuation or plastic deformation[3,4].

We characterize the surfaces of impacted sites using ex-situ AFM to determine the extent of damage-induced plasticity. Representative AFM height profiles for contact areas corresponding to $v_i = 407$ ms$^{-1}$, 414 ms$^{-1}$, and 515 ms$^{-1}$ (impact tests highlighted in Fig. 2b, c) feature shallow craters with material pile-up (Fig. 3a, b), confirming that the MA-BCP film dissipates energy by plastic deformation. We consistently observe these deformed surfaces with the depth of the crater ($\delta$) increasing with impact velocity (Supplementary Fig. 3a,d), thus indicating that the energy dissipated by the film increases with kinetic energy input. Additionally, AFM phase images of the selected contact areas (Fig. 3c–e) show evidence of brittle failure in the form of cracks around the perimeter of the impacted site, which we attribute to the presence of the stiff PS domains. Zoomed insets taken near the center of the contact areas (white boxes in Fig. 3c–e) surprisingly show that the phase-separated morphology of the BCP microstructure is retained at the base of the crater, although the periodicity and long-range ordering changed compared to the pristine film (Supplementary Fig. 4a–d).

To supplement AFM measurements, we image impacted sites using fluorescence microscopy (FM) to visualize any mechanophore activation. FM images of the contact surfaces shown in Fig. 3c–e exhibit a fluorescence signal around the perimeter of the craters (Fig. 3f–h). The formation of pile-up involves particle motion from the center of the contact area towards the edges[29]. We surmise that this motion was sufficient to rupture and consequently activate mechanophores along the contact perimeter. Interestingly, the intensity of the fluorescence signal increases with impact velocity (Supplementary Fig. 3a,c), which is nominally consistent with an increase in the concentration of mechanophores activated[17] and confirms that energy dissipation scales with kinetic energy input. The contact radius ($a$) measured by AFM and FM for varying impact velocities is within 2% error (Supplementary Fig. 3e and Supplementary Table 1). The strong agreement between both measurements suggests that the MA mechanophores act as efficient damage sensors in the MA-BCP material to report local damage.

Notably, the FM measurements also reveal mechanophore activation well below the surface of the impacted site for the faster impact velocities investigated ($v_i = 397$ ms$^{-1}$ to 515 ms$^{-1}$) (Supplementary Fig. 5a,b). By visualizing the FM results in 3D (Fig. 4a), we find that the subsurface mechanophore-activated volume resembles a cone. A 2D slice of this 3D projection illustrates that mechanophore activation occurs on the lateral surface of the cone and tapers to the apex located several microns below the permanently deformed film surface (Fig. 4b).

The mechanophore-activated volume bears a striking resemblance to a Mach cone. The formation of a Mach cone typically involves a disturbance penetrating a medium at a velocity that exceeds the shear and/or longitudinal wave velocity of that medium[1,30–33]. Acoustic waves radially expand into the medium behind the propagating disturbance and consequently superimpose to generate either one or two nested cones (shear or longitudinal

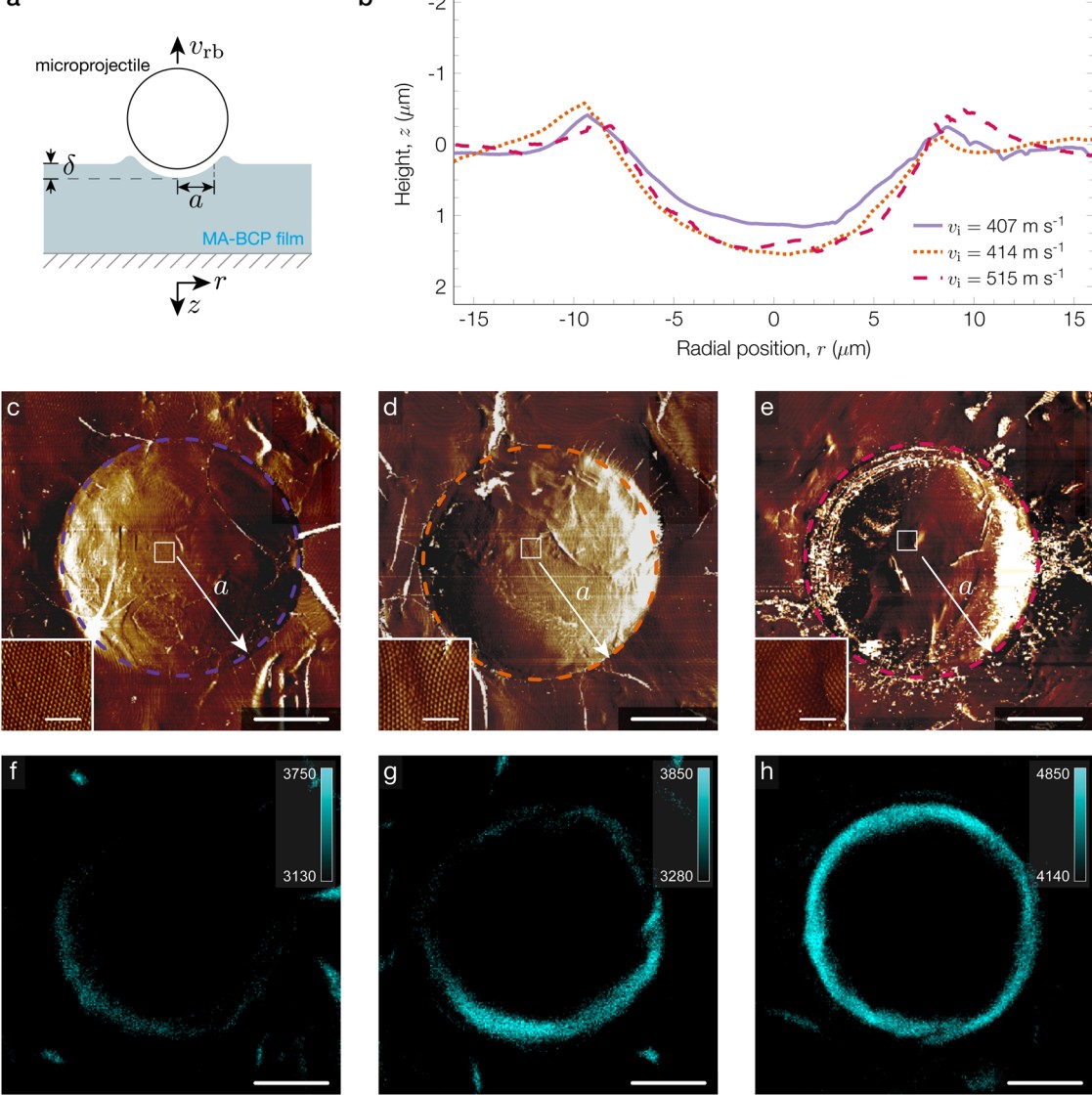

**Fig. 3 | Ex-situ surface characterization of impacted sites on MA-BCP films.**
**a** Schematic of the local deformation induced by the microprojectile on the MA-BCP film with a contact radius ($a$) and depth ($\delta$). **b** AFM line scans of three different impacted sites highlighted in Fig. 2b, c. **c–e** AFM phase images corresponding to impact velocities of 407 ms$^{-1}$, 414 ms$^{-1}$, and 515 ms$^{-1}$, respectively. The dashed line indicates the perimeter of the contact area and the scale bar represents 5 $\mu$m. The inset shows a magnified image of the BCP nanostructure taken near the center of the impacted site; the scale bar represents 500 nm. **f–h** FM images taken on the film surface ($z = 0$ plane) and corresponding to the contact areas **c–e**, respectively. The scale bar represents 5 $\mu$m.

Mach cone, respectively)[30–33]. The leading edges of the cone(s) are shockwave fronts characterized by significant particle motion, whereas the areas outside and within the cone(s) are relatively undisturbed[31,32]. Similarly, we observe mechanophore activation localized on the lateral surface of a cone-shape. The localized motion that occurs along the shock fronts (i.e., cone walls) could, in principle, rupture and thus activate mechanophores in the shape of the hollow cone observed in FM measurements.

However, unlike examples of Mach cone formation involving penetration into the material, our impact experiments barely deform the MA-BCP film ($\delta \le 1.8\,\mu$m). To provide insight into the wave propagation and particle motion in our material in response to impulsive deformation, we computationally model the impact event using finite element analysis (FEA). The FEA simulations suggest that the energy transfer at the microprojectile-MA-BCP interface generates a moving source that briefly propagates into the material at a velocity approaching the microprojectile impact velocity and exceeding the shear wave velocity of the material (Supplementary Fig. 6a). This

high-velocity source induces significant subsurface motion contained within a V-shape, in 2D, that is geometrically comparable to the fluorescence behavior captured in experiments (Fig. 4c and Supplementary Fig. 6b) and indirectly points to a high-speed particle source capable of driving shear Mach cone formation (Supplementary Fig. 6c). FM measurements corroborate shear Mach cone formation as they consistently show only one cone per impact event[30–33]. Taken together, we can make two comments about these results. First, our LIPIT experiments can cause Mach cone formation in the MA-BCP in a manner that is analogous to a focused ultrasound experiment, which remotely generates a subsurface mechanical moving source of vibration capable of creating a Mach cone in soft materials[30]. Second, shear Mach cone formation indicates that the microprojectile impact induces subsurface velocities faster than the shear wave velocity, but slower than the longitudinal wave velocity of the material[30–33].

The shear wave velocity ($C_S$) of the MA-BCP material is related to the Mach cone angle ($\alpha$) and microprojectile impact velocity according

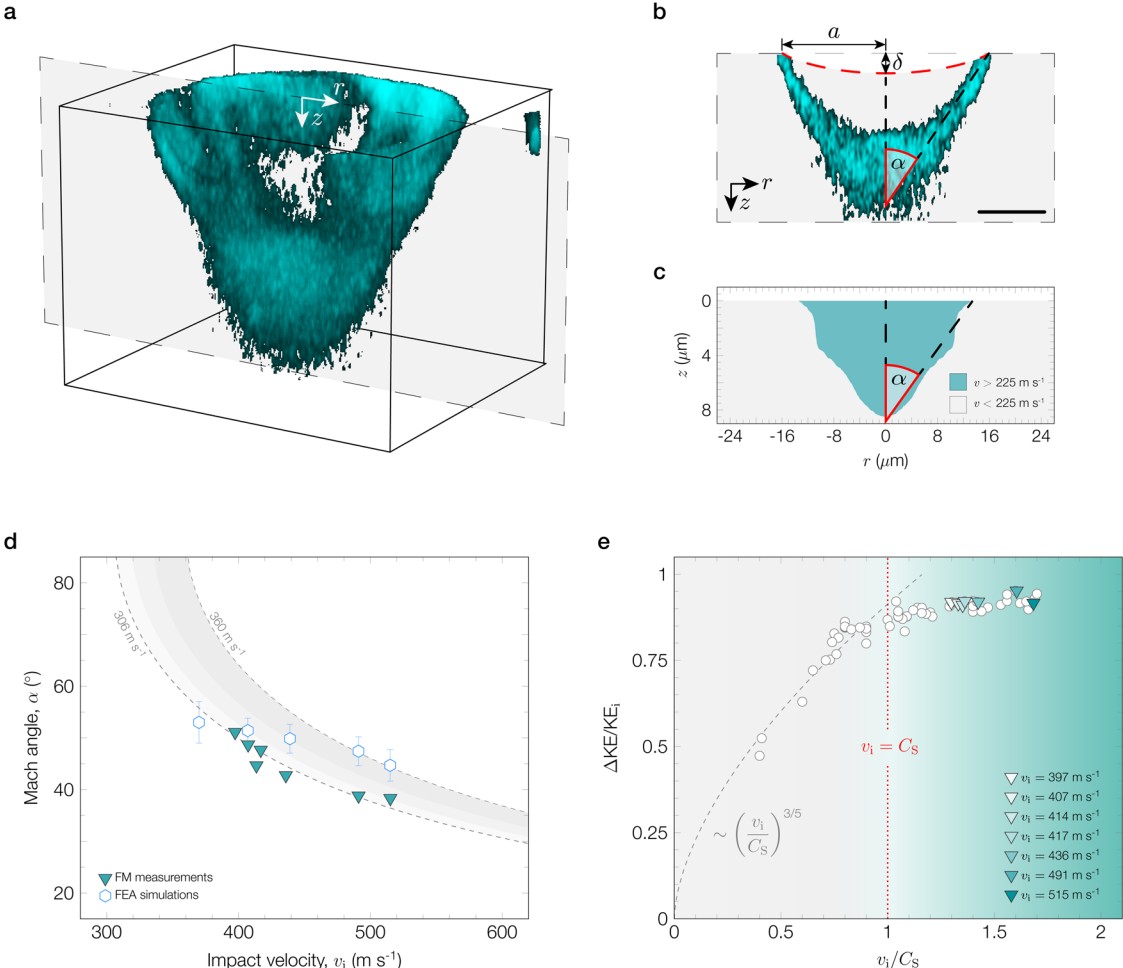

**Fig. 4 | Visualization and simulation of shock deformation within the bulk of the MA-BCP films. a** 3D projection of the mechanophore-activated volume below the impacted site corresponding to the impact velocity case, $v_i = 414\,\mathrm{ms^{-1}}$. **b** A 2D slice of the 3D projection, from **a** shows that the activated volume resembles a Mach cone. The dashed red line indicates the deformed film surface as measured by AFM, and the scale bar represents 5 $\mu$m. **c** Velocity contour map of the BCP material extracted from an FEA simulation of the microballistic impact experiment, with $v_i = 407\,\mathrm{ms^{-1}}$. **d** The Mach cone angle ($\alpha$) extracted from FM measurements and FEA simulations versus impact velocity ($v_i$). Data from the FEA simulations are

presented as mean values ± SD. The gray curves are fits to the data that extrapolate the shear wave velocity of the MA-BCP material according to the Mach-cone-angle relationship Eq. (1). **e** Kinetic energy loss parameter ($\Delta\mathrm{KE/KE_i}$) replotted as a function of impact velocity normalized by the shear wave velocity extracted from FM measurements ($v_i/C_S$). Impact cases in the subsonic regime ($v_i < C_S$) are fit by the impact model in Eq. (2). Mach cones are only observed in FM measurements for impacts in the intersonic regime ($C_S < v_i < C_L$) with representative cases highlighted by the triangle symbols.

to the Mach-cone-angle relationship[31–33],

$$\alpha = \sin^{-1}\left(\frac{C_S}{v_i}\right) \qquad (1)$$

For polymers, the shear wave velocity is typically strain rate dependent due to viscoelasticity. Based on our estimates and supported by simulations, the penetration strain rate is nearly constant for all impact events ($\approx 10^8\,\mathrm{s^{-1}}$), and thus the shear wave velocity of the MA-BCP film is expected to be constant. Accordingly, the mechanophore-activated cone angles measured for varying impact velocities are found to be consistent with Eq. (1) with an excellent fit corresponding to $C_S = 306\,\mathrm{ms^{-1}}$ (Fig. 4d). The cone angles extracted from FEA simulations are in close agreement with FM measurements, but display a slight increase in the shear wave velocity with increasing impact velocity. We attribute this slight difference in wave velocities to the hyperelastic homogeneous material assumption and increased strain hardening of the material observed in the higher impact velocity simulations.

As an additional validation of the shear wave velocity obtained from FM measurements, we perform Brillouin light scattering (BLS) measurements on the MA-BCP films. Our film consists of stiff PS nanospheres in a compliant PIB matrix and, therefore, has a structure comparable to a 2D Yukawa solid comprising of rigid microspheres suspended in a plasma medium. In the 2D Yukawa solid, the formation of Mach cones occurs via the center-of-mass displacement of microspheres rather than the deformation of individual microspheres[31,32]. We assume a similar mechanism for Mach cone formation for our materials; the deformation of the MA-BCP material predominantly occurs in the PIB matrix rather than the stiffer PS nanospheres. Since the BLS technique considers a nanostructured material as a homogeneous medium, we extrapolate a shear wave velocity for the PIB phase using an effective medium approach[34] and obtain $C_S = 338\,\mathrm{ms^{-1}}$, which is in close agreement with our FM measurements. The BLS measurements are performed at a slightly higher frequency ($10^9\,\mathrm{s^{-1}}$) compared to impact experiments ($10^8\,\mathrm{s^{-1}}$), which may account for the slight difference in strain-rate-dependent shear wave velocity obtained by each technique.

Kinetic energy transfer calculations provide our final support for the formation of shear Mach cones in the MA-BCP film. As shown in Fig. 4e, we compare the kinetic energy loss parameter with the dimensionless impact velocity, defined as the ratio of the micro-projectile impact velocity versus the shear wave velocity obtained from FM measurements ($v_i/C_S$). The case where $v_i = C_S = 306\ \text{ms}^{-1}$ clearly demarcates the data into two regimes. The first regime, where the impacts are slower than the shear wave velocity of the material ($v_i < C_S$), is defined as the subsonic regime. Consistent with the theory of Mach cone formation, we did not observe any Mach cones from FM measurements for subsonic impacts. AFM measurements of subsonic impacted sites indicate minimal permanent deformation of the film surface without any material pile-up. According to impact models for shallow deformation, the kinetic energy loss parameter scales with the impact velocity and shear wave velocity as[35],

$$\frac{\Delta \text{KE}}{\text{KE}_i} = A_1 \left(\frac{v_i}{C_s}\right)^{3/5} \qquad (2)$$

where $A_1$ is a parameter that is related to the elasticity of the projectile and the impacted material. Using the shear wave velocity extracted from FM measurements, we find that Eq. (2) produces a strong fit to the subsonic data (Fig. 4e).

The subsonic regime then transitions into the intersonic regime where impact velocities exceed the shear wave velocity, but are below the longitudinal wave velocity ($C_S < v_i < C_L$) of the material. In this second regime, we consistently observe Mach cone formation from FM measurements (Fig. 4e). Since mechanophore activation requires the rupture of covalent bonds in the MA molecule, this implies that local bond rupture and yielding occur at these impact velocities. Furthermore, the observation of mechanophore activation at the interface of hard and soft domains provides visible evidence of the shock-attenuating properties of nanoscale materials with spatially varying viscoelastic properties[36]. The strategic placement of hard nanostructures in a soft matrix can act to disrupt, deflect, or diminish the shockwave, analogous to the use of solid obstacles for shockwave attenuation in air[37], suggesting significant energy dissipated via Mach cone formation[1]. While some energy is dissipated by mechanophore bond scission, a recent molecular dynamics study suggests that this is an insignificant contribution to the overall energy dissipation[38].

In high-velocity impacts exhibiting material pile-up, plastic deformation is conventionally presumed to be the dominant energy dissipation mechanism, whereas shockwave attenuation is considered negligible[3,4]. However, this present study captures both mechanisms for this MA-BCP system, thereby uncovering important insights on how solid materials dissipate energy during an impulsive event. By integrating mechanophores into various geometries and materials systems, this material platform provides a promising approach to investigate the roles of interfaces, phase changes, and chemomechanical changes in shockwave interactions in materials.

In summary, we incorporated mechanochemistry in microballistic studies to reveal insights into the chemical and mechanical responses of a material to high-strain-rate events ($10^8\ \text{s}^{-1}$). Our mechanophore block copolymer system revealed that both acoustic wave attenuation and plastic deformation act as energy dissipation mechanisms during a microballistic impact, which bridges viewpoints from both impact physics and shockwave physics communities. We envisage that investigating other mechanophore-functionalized materials in this integrated platform can enable further understanding of other impulsive events, such as mild traumatic brain injuries ($10\ \text{s}^{-1}$), cold spray additive manufacturing ($10^7\ \text{s}^{-1}$), and hypervelocity impacts in space ($10^{12}\ \text{s}^{-1}$)[17].

## Methods

### Materials
The mechanophore containing polyisobutylene-*b*-polystyrene (PIB-*b*-PS) diblock copolymer is synthesized by coupling the PIB block and PS block via a Diels-Alder reaction[24]. The PIB block is first synthesized by quasi-living cationic polymerization and then converted to maleimide end-functionalized PIB via a series of post-polymerization reactions. The anthracene end-functionalized PS block is synthesized separately via thermal RAFT using an anthracene functional DMP RAFT agent. The maleimide functional PIB and anthracene functional PS are then coupled via the Diels Alder reaction to yield the maleimide-anthracene-functionalized block copolymer (MA-BCP). The detailed synthesis is described in the Supplementary Methods. The MA-BCP has $M_n = 106\ \text{kg mol}^{-1}$ and $M_w = 121\ \text{kgmol}^{-1}$ ($M_w/M_n = 1.14$) as determined by gel-permeation chromatography measurements. The PIB volume fraction is 0.57 as calculated from nuclear magnetic resonance spectroscopy. The morphology of the self-assembled block copolymer film (i.e., lamellar, cylindrical, or spherical) is controlled by film preparation and annealing conditions.

### Film preparation
The MA-BCP material is received in powder form. A thin film is prepared by first dissolving 50 mg of the MA-BCP powder in 0.8 mL of dry toluene. The MA-BCP solution is then filtered using a 0.45 $\mu$m Teflon filter (28143–924, VWR). Next, 0.6 mL of the filtered solution is deposited on a 24 mm × 40 mm glass cover slip (2975–244, Corning) and enclosed within a glass cell. The solution slowly evaporates at room temperature for 24 hours. The dried film is approximately 50 $\mu$m thick, as measured by optical profilometry (Zygo NewVIEW 7300, AMETEK Inc.).

### Small-angle X-ray scattering measurements
Small-angle X-ray scattering (SAXS) measurements are made at beamline 11-BM (CMS) at the National Synchrotron Light Source II at Brookhaven National Laboratory with a 0.92 Å (13.5 keV) X-ray source. The 1D SAXS profiles are obtained by azimuthally averaging the 2D data and normalizing background scattering via an exponential decay function. Images are plotted as intensity ($I$) versus scattering vector ($Q = (4\pi/\lambda)\sin\theta$), where $\lambda$ is the wavelength of the incident X-ray beam and $2\theta$ is the scattering angle (Supplementary Fig. 1).

### Impact testing
Microballistic impact experiments are performed on the MA-BCP film using our laser-induced particle impact test (LIPIT) platform[11]. A pulsed diode-pumped solid-state IR laser (Flare NX, wavelength = 1030 nm, pulse length = 1.5 ns, Coherent Inc.), with a 20 × objective (Olympus Plan Fluorite Objective, Thorlabs Inc.), expands the launch pad via laser ablation. The launch pad comprises of a glass coverslip coated with ≈ 30 nm thick gold and covered with an elastomeric (crosslinked polydimethylsiloxane, PDMS, Dow Corning Sylgard 184) layer ≈ 20–50 $\mu$m thick depending on the desired impact velocity. The expansion of the launch pad accelerates a monodisperse silica microprojectile (20 $\mu$m diameter, SiO2-R-20.0, microParticles GmbH) towards the MA-BCP film surface (Supplementary Fig. 2a).

An ultrafast camera (SIMD12, Specialized Imaging Ltd.) captures the trajectory of the microprojectile with 30 ns of exposure time per frame and interframe time ranging from 150 ns to 450 ns, depending on the impact velocity. The camera has a maximum of 12 frames, and the error of the camera is ± 3 ns. The synchronization of the laser ablation process and image acquisition is achieved via digital triggers modulated using a digital waveform generator (NI-9402, National Instruments). Each image collected is subjected to background subtraction and thresholding to improve image quality using ImageJ. The

velocity is calculated by dividing the distance traveled by the micro-projectile between two consecutive frames by the time between frames (sum of exposure and interframe times). The energy dissipated by the film is quantified by the kinetic energy loss parameter ($\Delta$KE/KE$_i$)[3],

$$\frac{\Delta\mathrm{KE}}{\mathrm{KE_i}} = \frac{\frac{1}{2}m_\mathrm{p}v_\mathrm{i}^2 - \frac{1}{2}m_\mathrm{p}v_\mathrm{rb}^2}{\frac{1}{2}m_\mathrm{p}v_\mathrm{i}^2} = 1 - \frac{v_\mathrm{rb}^2}{v_\mathrm{i}^2} \tag{3}$$

which is a function of the microprojectile mass ($m_\mathrm{p}$), impact velocity ($v_\mathrm{i}$), and rebound velocity ($v_\mathrm{rb}$).

The penetration strain rate[5,28] is approximated as $\dot{\varepsilon} = v_\mathrm{i}/\delta$, where $\delta$ is the depth of the impacted site. The strain rates are estimated to be on the order of $10^8\,\mathrm{s}^{-1}$ for all impact tests (Supplementary Table 1). The contact duration between the projectile and film ($t_\mathrm{tot}$) is calculated by[29],

$$t_\mathrm{tot} = \frac{\pi}{4}\left(1 + 1.2\frac{v_\mathrm{rb}}{v_\mathrm{i}}\right)\frac{\delta}{v_\mathrm{i}} \tag{4}$$

Since the depth of the impacted site increases with impact velocity (Supplementary Fig. 3d and Supplementary Table 1), the contact times are calculated for a range of depths and are estimated to remain below 20 ns for all tests (Supplementary Fig. 2b).

By comparison, the time required for one acoustic wave to reflect across the film is calculated as $T_1 = 2h/C_\mathrm{S}$, where $h$ is the film thickness ($\approx 50\,\mu$m) and $C_\mathrm{S}$ is the shear wave velocity of the material. The number of wave reflections that can occur during an impact event is thereby calculated as $N_\mathrm{wr} = t_\mathrm{tot}/T_1$. The high-strain-rate $C_\mathrm{S}$[39] and calculated $T_1$ of the PS phase are 1150 ms$^{-1}$ and 87 ns, respectively. The $C_\mathrm{S}$ extracted from FM measurements and calculated $T_1$ of the PIB phase are 306 ms$^{-1}$ and 327 ns, respectively. As shown in Supplementary Fig. 2c, we estimate that no shear wave reflections occur at any impact velocity for either PS or PIB phase.

## Fluorescence microscopy measurements

Fluorescence measurements (FM) of the impacted sites are imaged within two hours of LIPIT testing using a Nikon Ti2 eclipse inverted microscope. Images are collected at 460 nm ± 25 nm emission upon 350 ± 25 nm excitation at 60 × magnification and 1.4 numerical aperture. For each impacted site, $z$-stacked images are taken from the film surface down to several microns into the film with a $z$-step of 0.3 $\mu$m. FM images corresponding to AFM surface measurements (Fig. 3f–h and Supplementary Fig. 3a) are taken on the film surface (i.e., $z = 0$ plane). Using ImageJ, radial intensity profiles are taken from the center of the impacted site to the exterior at every 10° for every $z$-plane. Representative radial intensity profiles at $\theta = 0°$, 10°, 20° taken on the film surface for an intersonic impact case ($v_\mathrm{i} = 414$ m s$^{-1}$) show a peak intensity near the perimeter of the impacted site (indicated by the black arrow in Supplementary Fig. 3b). The maximum fluorescence intensity ($I_\mathrm{max}$) and corresponding radial location ($r_\mathrm{max}$) are recorded for the 36-line profiles and then averaged using a custom Matlab script. The averaged $I_\mathrm{max}$ on the film surface is plotted in Supplementary Fig. 3c. The averaged $r_\mathrm{max}$ on the film surface is defined as the contact radius of the impacted site and plotted in Supplementary Fig. 3e. 3D projections of $z$-stacked images and 2D slices of 3D projections are obtained using ImageJ. Intersonic impact cases ($v_\mathrm{i} = 397$ ms$^{-1}$ to 515 ms$^{-1}$) exhibit subsurface mechanophore activation in a cone shape (Fig. 4a). As shown in Supplementary Fig. 5b, the cone angle ($\alpha$) of the mechanophore-activated volume is extracted from the slope ($m$) of the best-fit trendline for $r_\mathrm{max}$ versus $z$ according to $\alpha = \tan^{-1}(m)$.

## Atomic force microscopy measurements

Atomic force microscopy (AFM) is performed in tapping mode on a Bruker Dimension Icon. The spring constant of the cantilever is 6 Nm$^{-1}$ with a resonant frequency of 150 kHz. Height profiles are obtained by taking a line scan across the center of the impacted site. The contact depth of the impacted site ($\delta$) is defined as the minimum point in the height profile. The contact radius ($a$) is defined as the $x$-intercepts in the height profile (Fig. 3a). Fast Fourier transformation (FFT) and azimuthal integration of the phase images are performed in ImageJ (Supplementary Fig. 4a–d).

## Finite element analysis modeling

Experimental findings are supplemented with explicit Finite Element Analysis (FEA) of a homogeneous 2D material impacted at velocities spanning 370 ms$^{-1}$ to 515 ms$^{-1}$. Simulations are conducted using the hydrodynamic FEA code ALE3D[40], and axisymmetric boundary conditions are used with 2D reduced-order quadrilateral elements. The model consists of a polymer film (50 × 640 $\mu$m$^2$) on a silica substrate (270 × 640 $\mu$m$^2$) impacted by a silica sphere (20 $\mu$m diameter). The only reflecting boundaries are the interfaces between the impactor and polymer and the polymer and substrate. All other boundaries are approximated as semi-infinite mediums by using non-reflecting and pressure outflow boundaries in ALE3D.

Simulating a phase-separated co-polymer with hard spherical clusters at the nano-scale is a difficult task. The PIB phase behaves hyperelastically[41] and can undergo several hundred percent strain before failure, while the PS phase exhibits a high-strength quasi-brittle behavior[42]. For this work, we chose an idealized homogeneous hyperelastic material with material properties fit from the observed experimental data. We use a volumetrically uncoupled Mooney-Rivlin model based on the left Cauchy-Green tensor (**B**),

$$\mathbf{B} = \mathbf{F}^\mathrm{T}\mathbf{F} \tag{5}$$

where (**F**) is the deformation gradient. The volumetric response was removed by taking the deviatoric parts of the scaled strain $\left(\tilde{\mathbf{B}}\right)$,

$$\tilde{\mathbf{B}} = \frac{\mathbf{B}}{\det(\mathbf{B})^{\frac{1}{3}}} \tag{6}$$

$$\tilde{\mathbf{B}}' = \tilde{\mathbf{B}} - \frac{1}{3}\mathrm{tr}\left(\tilde{\mathbf{B}}\right) \tag{7}$$

The strain energy as a function of strain (**B**), volume change (**J**) and internal energy (**E**) is given as:

$$\mathbf{W}(\mathbf{B},\mathbf{J},\mathbf{E}) = \mathbf{W}(\mathbf{J},\mathbf{E}) + \frac{1}{2}\mu_0\left[\omega\left[\mathrm{tr}(\tilde{\mathbf{B}}) - 3\right] + (1+\omega)\left[\mathrm{tr}(\tilde{\mathbf{B}}^{-1}) - 3\right]\right] \tag{8}$$

where $\mu_0$ is the initial shear modulus, $\omega$ is a shaping factor, and **W(J,E)** is determined independently by the equation of state (EOS). The deviatoric part of the Cauchy stress tensor can now be defined by differentiating the deviatoric part of the strain energy function with respect to the deviatoric left Cauchy-Green strain tensor,

$$\boldsymbol{\sigma}' = \frac{\mu_0}{\mathbf{J}}\left[\omega\tilde{\mathbf{B}}' - (1-\omega)\left((\tilde{\mathbf{B}}^2)' - \mathrm{tr}(\mathbf{B})\mathbf{B}'\right)\right] \tag{9}$$

where $\mathbf{J} = \sqrt{\det(\mathbf{B})}$ is the Jacobian and is equal to the relative volume. The shaping factor is assumed to be 0.8. The initial shear modulus is determined from the relationship $\mu_0 = \rho C_\mathrm{S}^2$, where $\rho$ is the film density and $C_\mathrm{S}$ is the shear wave velocity. Assuming a density of 1000 kgm$^{-3}$ and a wave velocity of 306 ms$^{-1}$ as found through FM measurements, the initial shear modulus is $\approx 93.5$ MPa. The volumetric response is governed by a linear pressure-volume response only requiring the initial bulk modulus. Pressure-volume data[43] for PIB gives a bulk modulus of $\approx 3$ GPa and is used in this study.

The software VisIt[44] is used for visualization of the stresses, strains, and velocities imparted into the film by the projectile. An

element size of 0.5 μm is selected for the projectile mesh. The polymer and substrate mesh consists of a fine region (0.5 μm element size) spanning 60 μm radius around the projectile and a coarse region spanning the remaining 280 μm. The mesh in the coarse region is biased so that the element size is initialized at the fine region mesh size and expands by a factor of 5 at the outermost boundaries. In all, the model consists of ≈ 60,000 elements.

Two custom variables are created in ALE3D for visualization purposes: (1) the volumetric strain rate and (2) the nodal peak velocity achieved during the impact event. The volumetric strain rate allows for the visualization of the pressure and shear waves as they propagate through the medium (Supplementary Fig. 6a). The pressure wave travels much faster than the projectile and shear waves. By tracking the pressure wave front location at subsequent 20 μs time intervals, the average pressure wave velocity ($C_L$) is found to be ≈ 1650 ms$^{-1}$. It is observed that the projectile is initially faster than the shear wave, as the wave is not initially apparent around the time of impact. At 40 μs, the shear wave starts to separate itself from the projectile as the projectile slows. By tracking the shear wave front vs. time, the shear wave velocity ($C_S$) is found to be ≈ 321 ms$^{-1}$.

The nodal peak velocity illustrates that cone-like areas extending far below the surface experience significant velocities, indicating a possible mechanism of Mach cone formation. For a qualitative comparison, the threshold value of 225 ms$^{-1}$ was chosen to create the simulated velocity cones (Supplementary Fig. 6b). The formation of a Mach cone requires particle velocities exceeding the wave speed (in this case, shear wave speed) of the medium in question (Supplementary Fig. 6c). It was observed that high velocities can extend well beneath the material surface. The angle of these cone-like areas grows larger at increased impact velocities in a manner like that observed experimentally. Note that these simulations do not account for the true heterogeneous nature of the phase-separated material, but act to qualitatively demonstrate the shear wave velocity versus impact velocity relationship and to illustrate the propagation of particle velocities well underneath the projectile-target interface.

**Acoustic measurements**
Brillouin light scattering (BLS) is performed under ambient conditions using a 532 nm Verdi V2 DPSS green laser (Coherent Corp.) as the probing light source. The MA-BCP film is cast on a silicon wafer and then directly used to collect the BLS spectrum in the emulated platelet geometry[45]. From the BLS measurements shown in Supplementary Fig. 7, we obtain the longitudinal wave speed, $C_L$ ≈ 2420 ms$^{-1}$. Using the relationship, $M = \rho C_L^2$, the elastic constant obtained from this film is $M_{BCP} = 5.85$ GPa assuming a film density of $\rho_{BCP}$ ≈ 1000 kgm$^{-3}$.

The longitudinal modulus ($M$) for a two-component composite material with a random arrangement of spheres (isotropic) can be estimated based on a simple rule of mixtures[34,46],

$$\frac{1}{M} = \frac{\phi_1}{M_1} + \frac{\phi_2}{M_2} \tag{10}$$

where $\phi_1$ and $\phi_2 = 1 - \phi_1$ are the volume fractions of components 1 and 2, respectively. The parameters $M_1$ and $M_2$ are the longitudinal moduli for 1 and 2, respectively. Using effective medium theory, with the relationship of the elastic constants for an isotropic material ($M = 2\mu(1 - \nu)/(1 - 2\nu)$), we can estimate the shear modulus of the PIB phase,

$$\mu_{PIB} = \frac{1}{2} \left( \frac{\phi_{PIB}}{1/M_{BCP} - \phi_{PS}/M_{PS}} \right) \left( \frac{1 - 2\nu_{PIB}}{1 - \nu_{PIB}} \right) \tag{11}$$

Assuming $\phi_{PIB} = 0.57$, $\phi_{PS} = 1 - \phi_{PIB} = 0.43$, and $\nu_{PIB} = 0.49$. The acoustic wave speed of PS[39] is assumed to be $C_{L,PS} = 2400$ m s$^{-1}$. This translates to $M_{PS}$ ≈ 6.05 GPa ($\rho_{PS} = 1050$ kgm$^{-3}$). Substituting these values

into Eq. (11), we obtain $\mu_{PIB}$ ≈ 112 MPa. Since $\mu = \rho C_S^2$, we calculate that $C_{S,PIB}$ ≈ 338 ms$^{-1}$ from BLS (assuming $\rho_{PIB} = 980$ kgm$^{-3}$), which is in close agreement with our impact studies where we estimate $C_{S,PIB}$ ≈ 306 ms$^{-1}$.

## Data availability
The data that support the findings of this study are available from the corresponding author upon request.

## Code availability
The code that support the findings of this study are available from the corresponding author upon request.

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

## Acknowledgements

This research used beamline 11-BM (CMS) of the National Synchrotron Light Source II, a US Department of Energy (DOE) Office of Science User Facility operated for the DOE Office of Science by Brookhaven National Laboratory under contract No. DE-SC0012704. We thank Dr. Christopher Stafford for access to the atomic force microscope. Certain instruments and materials were identified in this paper to adequately specify the experimental details. Such identification does not imply recommendation by the National Institute of Standards and Technology (NIST), nor does it imply that the materials are necessarily the best available for the purpose. This work was supported by the NIST National Research Council (NRC) postdoctoral fellowship (P.J.C.), the NSF through the CAREER Award (1945092) (P.V., Y.C.S.), and the US Army Engineer Research and Development Center (ERDC) under ERDC BAA 18-0500 and BAA 20-0110 "Multifunctional Materials to Address Military Engineering" executed under Contract Nos. W912HZ-18-C-0022 and W912HZ-21-C-0029 (P.V., Y.C.S., K.D.M., S.E.M.). Permission to publish was granted by Director, Geotechnical and Structures Laboratory.

## Author contributions

P.J.C. and E.P.C. conceived the idea. P.J.C., K.M.E., T.L.T., C.L.S., S.E.M., Y.C.S., and E.P.C directed the research. P.J.C., K.D.M., P.V., and L.H. performed the experiments. A.L.B. conducted the computational simulations. P.J.C., A.L.B., and E.P.C. wrote the manuscript. All authors participated in editing the manuscript.

## Competing interests

The authors declare no competing interests.
