## [Peer Review File · Nature Communications]

Mechanochemically Responsive Polymer Enables Shockwave VisualizationREVIEWER COMMENTS

Reviewer #1 (Remarks to the Author):

Review of Centellas et al.

This manuscript describes the use of mechanophores to visualize shockwave dissipation in block copolymers, and distinguish between two mechanisms and regimes. The idea is novel and the experiments are well-designed and performed. This work should be of significance to the broader community interested in energy dissipation. Overall I think this is suitable for publication. I have a few comments/questions:

1. Is there any role of the impedance mismatch in PIB/PS on these results. Presumably since they are all the same block polymer, the dissipation is not related to the specific chemistry.
2. Would the presence of the mechanophore within the PS or PIB domains change how they break in response to the shockwave? Is it important to be sitting at the interface of a hard/soft material? If the spheres undergo a center of mass displacement, perhaps a mechanophore sitting in PS would not respond.
3. Is the energy dissipated in breaking the mechanophore a significant contribution to the overall dissipation?
4. Do the authors anticipate this method will work in an entirely soft material, like PDMS crosslinked with mechanophores?

Reviewer #2 (Remarks to the Author):

This manuscript demonstrates a novel methodology to visualize shockwaves induced by microballistic impact. To visualize the damaged area, the authors used an anthracene-maleimide mechanophore that activates luminescence, incorporated at the junction points of a block copolymer composed of soft polyisobutylene and hard polystyrene. Upon impact by a tiny silica microprojectile traveling at a velocity exceeding the shear wave velocity, a Mach cone-shaped damage pattern is induced in the polymer containing the mechanophore. This mechanophore activation is visualized three-dimensionally using laser scanning confocal microscopy (LSCM). Additionally, finite element analysis simulations provide further insight into how the shockwave propagates and induces Mach cone formation. The simulated Mach cone angles closely align with the LSCM results. This study, which focuses on high-strain rate mechanical induction, has broad implications for various research fields since previous studies on mechanochromic mechanophores have only examined slower strain rates. Furthermore, the results indicate that shockwave attenuation is as important as plastic deformation. However, the reviewer has several concerns regarding the experimental data, theoretical data, and the descriptions provided. Therefore, I recommend that the manuscript be considered for publication in Nature

Communications after the following points are fully addressed:

1. The waves propagating inside the polymer affect the formation of the Mach cone and its angle. To what extent do the velocities of the waves propagating inside the polymer change with the speed of the silica microprojectile?
2. How is the mechanophore activated in response to a shockwave? Is it due to polymer chain elongation, direct detection of the shockwave, or other mechanisms? To gain more insight into the activation mechanism, the same experiment should be performed with a polymer in which the mechanophores are physically doped.
3. How did you calculate and obtain the averaged maximum fluorescence I_{max} ? Is I_{max} derived from the intensity at a specific radius? Please explain the definition of I_{max} in more detail. Additionally, a 2D or 3D fluorescence intensity map should be included.
4. The authors mention conducting Brillouin light scattering; however, the reviewer cannot find these results in the manuscript. Please include the data.
5. The authors explain that the polystyrene spheres in the copolymer attenuate the shockwave. To support this explanation, the same experiment should be conducted with polyisobutylene films in which the mechanophores are covalently introduced.
6. Are the LSCM images in Fig. 3f–h single z-step images or constructed from several z-steps? Please provide more information on how these images were prepared.

Minor Points:

What is the “grey box” mentioned in line 137 and Fig. 4?

The images in Fig. 3f–h are not “fluorescence images” but LSCM images. Please correct this.

Ref. 33 was published in 2004. Please verify the citation again.

Dear Referees,

We greatly appreciate your thoughtful suggestions and insightful questions, and we have revised the manuscript accordingly. The edits to the manuscript and methods are highlighted in red. Please see below for responses to the specific questions.

Best regards,
Edwin P. Chan
Project Leader, Fundamentals of Polymer Mechanics
Materials Science and Engineering Division, NIST

Referee 1:

- Q1. Is there any role of the impedance mismatch in PIB/PS on these results. Presumably since they are all the same block polymer, the dissipation is not related to the specific chemistry.

As mentioned in the text, Mach cones have been reported in both heterogeneous materials, such as Yukawa solids (Ref. 36-37), and homogeneous materials, such as tissue-mimicking-phantom (Ref. 35) and PMMA (Ref. 38), suggesting that the results discussed in our manuscript are not limited to specific polymer chemistries nor morphologies.

In heterogeneous materials, such as our phase-separated nanostructure, shock wave dissipation is expected due to the impedance mismatch at the PIB/PS interface (discussion on page 13). In homogeneous materials, such as viscoelastic PMMA (Ref. 38), non-uniform material stiffening in the vicinity of the Mach cone induces a transient heterogeneity in the material properties of the homogeneous matrix. We expect that this temporary heterogeneity dissipates shock wave energy in the homogeneous material similar to the mechanisms observed in heterogeneous materials.

These results are intriguing as they inspire new strategies to design nanostructures to dissipate impact energy. We are in the process of conducting a joint experimental and computational study on the energy dissipation behavior of various block copolymer morphologies to systematically investigate the role of impedance mismatch in these nanostructured materials.

- Q2. Would the presence of the mechanophore within the PS or PIB domains change how they break in response to the shockwave? Is it important to be sitting at the interface of a hard/soft material? If the spheres undergo a center of mass displacement, perhaps a mechanophore sitting in PS would not respond.

As our study required custom-designed materials, the placement of the mechanophore at the PIB/PS interphase was designed a priori to our impact studies. We agree with the Referee that had the mechanophores been situated within the PS domain, there might be insignificant mechanophore activation because the center-of-mass displacement of the PS spheres would not cause significant deformation of the PS chains.

This is a really interesting subject because there have been investigations on the effect of block copolymer composition and morphology on mechanophore activation (Ref. 7, 30), but, to the best of our knowledge, no study on the effect of mechanophore placement (within block versus at the interface of blocks) on activation. We intend to investigate this subject, but such a study would first require a series of large-scale atomistic simulations to help guide the mechanophore placement

along the polymer backbone, as it would be inefficient to investigate the optimal placement of the mechanophore empirically.

- Q3. Is the energy dissipated in breaking the mechanophore a significant contribution to the overall dissipation?

We previously explored this question via a multimillion-atom reactive molecular dynamics simulation to study the failure behavior of polystyrene experiencing a supersonic impact and found that the amount of energy dissipated due to bond scission ($\ll 0.1\%$) was negligible at impact velocities < 1 km/s. Although the simulations were on a different polymer system, the results are applicable here. We expect that the energy dissipated due to the rupture of the mechanophore bonds should be much smaller than the other energy dissipation mechanisms discussed in our response to Q1.

To clarify this point, we added the following statement (lines 254 - 256) and reference to the manuscript: "While some energy is dissipated by mechanophore bond scission, a recent molecular dynamics study suggests that this is an insignificant contribution to overall energy dissipation⁴²." Ref. 42: Bowman, A. L., Chan, E. P., Lawrimore, W. B., & Newman, J. K. Supersonic impact response of polymer thin films via large-scale atomistic simulations. *Nano Letters* **21**, 5991 - 5997 (2021).

- Q4. Do the authors anticipate this method will work in an entirely soft material, like PDMS crosslinked with mechanophores?

This is an excellent question by the Referee and one that we intend to explore in the future. Our high-strain-rate microballistic impact setup can be modified to access the appropriate impact velocity range to initiate Mach cone formation in entirely soft materials. For example, Bercoff, *et al.* (Ref. 35) studied Mach cones in soft-tissue-mimicking phantoms with shear and longitudinal acoustic wave velocities of 10 and 1500 m s⁻¹, respectively. Significant, localized particle motion is characteristic of Mach cone formation regardless of the matrix properties. Therefore, we hypothesize that this localized high-rate deformation should be sufficient to activate mechanophores even in a soft matrix. This hypothesis is supported by prior literature in polymer mechanochemistry, which reported activation in soft materials under varied high-strain-rate loading conditions, such as impact (Ref. 20) and compression of PDMS (Ref. 21).

Referee 2:

- Q5. The waves propagating inside the polymer affect the formation of the Mach cone and its angle. To what extent do the velocities of the waves propagating inside the polymer change with the speed of the silica microprojectile?

As discussed in the text and shown in Fig. 4d, the Mach cone angle is inversely proportional to the velocity of the silica microprojectile according to the Mach-cone-angle relationship (Eq.1).

- Q6. How is the mechanophore activated in response to a shockwave? Is it due to polymer chain elongation, direct detection of the shockwave, or other mechanisms? To gain more insight into the activation mechanism, the same experiment should be performed with a polymer in which the mechanophores are physically doped.

From both quasi-static compression (Ref. 26) and ultrasound (P. A. May, *et al.*, *ACS Macro Lett.*, 2016, 5, 177180) mechanical deformation experiments, it has been established that the polymer chains

must be attached to both ends of the mechanophore to cause activation. The mechanical deformation leads to the elongation of the polymer chains and mechanophores, and at sufficiently high stresses, this activates the maleimide-anthracene mechanophore. Stress transfer does not occur if only one end of the mechanophore is covalently connected to the polymeric network. Thus, physically doping the mechanophore into the block polymer would not lead to activation since the mechanophores would not bear any mechanical load.

To point out that there are different strain-rate dependent approaches to cause activation of the mechanophore via a chain deformation mechanism, we add the following statement (lines 90-93) to the manuscript: "As previously reported, there are various approaches to mechanically induce bond cleavage of mechanophores in polymeric materials, including ultrasonication^{28,31,32}, quasi-static tensile²⁷ and compression²⁶ testing, and high-strain-rate compression^{21,22} and shockwave loading¹⁹ experiments."

- Q7. How did you calculate and obtain the averaged maximum fluorescence I_{\max} ? Is I_{\max} derived from the intensity at a specific radius? Please explain the definition of I_{\max} in more detail. Additionally, a 2D or 3D fluorescence intensity map should be included.

Using ImageJ, radial intensity profiles are taken from the center of the impacted site to the exterior at every 10° for every z -plane. Figure R 1a shows representative radial intensity profiles at $\theta = 0^\circ, 10^\circ, 20^\circ$ taken on the film surface ($z = 0$ plane) for an intersonic impact experiment ($v_i = 414 \text{ m s}^{-1}$). The radial intensity profiles show a peak intensity near the perimeter of the impacted site (indicated by the black arrow in Fig. R 1a), which is also visualized by AFM and fluorescence images in Extended Data Fig. 3a. The maximum fluorescence intensity and corresponding radial location are defined as the I_{\max} and r_{\max} , respectively, for each radial intensity profile. Both values (r_{\max}, I_{\max}) are recorded for the 36-line profiles taken on a single z -plane and then averaged using a custom Matlab script. The averaged I_{\max} on the film surface ($z = 0$ plane) is plotted in Extended Data Fig. 3c. The corresponding averaged r_{\max} is plotted in Extended Data Fig. 3e and tabulated in Extended Data Table 1 for all intersonic impact experiments.

Representative radial intensity profiles at selected depths into the film ($z = 0 \mu\text{m}, 1.2 \mu\text{m},$ and $2.4 \mu\text{m}$) are shown in Fig. R 1b for the same experiment. These profiles show that the peak intensity occurs at smaller r_{\max} with increasing depth into the film, which is also illustrated by the 3D projection of z -stacked fluorescence images and the 2D slice of the 3D projection in Fig. 4a and 4b, respectively. To quantify the shape of the mechanochemically-activated volume, we plotted r_{\max} as a function of z as shown in Extended Data Fig. 5.

Figure R 1: Representative radial intensity profiles for an intersonic impact experiment ($v_i = 414 \text{ m s}^{-1}$). **a**, Profiles at $\theta = 0, 10, 20^\circ$ taken on the film surface ($z = 0$ plane). **b**, Select profiles at depths of $z = 0, 1.2,$ and $2.4 \mu\text{m}$ below the film surface.

We added Fig. R 1a and R 1b to Extended Data Fig. 3 and 5, respectively, and edited the Methods section to clarify the analysis of I_{max} .

- Q8. The authors mention conducting Brillouin light scattering; however, the reviewer cannot find these results in the manuscript. Please include the data.

We apologize for this error. We have now included the BLS spectra in the Extended Data section (Extended Data Fig. 7).

- Q9. The authors explain that the polystyrene spheres in the copolymer attenuate the shockwave. To support this explanation, the same experiment should be conducted with polyisobutylene films in which the mechanophores are covalently introduced.

We want to clarify that we are not implying that the polystyrene spheres in the block copolymer attenuate the shock wave. Instead, we attribute some of the energy dissipation to an acoustic impedance mismatch between the PIB/PS interfaces, which we discussed in our response in Q1. Specifically, impedance mismatch has been demonstrated to attenuate shock waves in various material systems (Ref. 40 and 41).

Our goal for this study was to demonstrate an approach for visualizing shock waves in soft materials, which has been difficult to do. This work was not intended to be a comprehensive analysis of shock wave propagation/attenuation in soft materials, which would require multiple studies that combine novel polymer synthesis efforts, microballistic impact experiments, and molecular dynamics simulations to understand how mechanophores couple with a polymer to attenuate shock waves. We appreciate this suggestion from the Referees, and we are in the process of designing new mechanophore-functionalized polymer systems to address these questions.

- Q10. Are the LSCM images in Fig. 3fh single z-step images or constructed from several z-steps? Please provide more information on how these images were prepared.

Figure 3f-h represent fluorescence images taken on the film surface ($z = 0$ plane). Only the 3D projection and the 2D slice of the 3D projection in Fig. 4a and 4b, respectively, are constructed from

several z -step images, as mentioned in our response to Q7. The caption for Fig. 3 and the text in the Methods section were edited to clarify that Fig. 3f-h correspond to a single z -plane image taken on the film surface.

Q11. Minor points:

What is the "grey box" mentioned in line 137 and Fig. 4?

The images in Fig. 3f-h are not "fluorescence images" but LSCM images. Please correct this.

Ref. 33 was published in 2004. Please verify the citation again.

The "grey box" refers to the insets in Fig. 3c-e and this description is mentioned in line 140 and Fig. 3 caption.

LSCM is a fluorescence microscopy technique (Bayguinov, P. O., Oakley, D. M, Shih, C.-C., Geanon, D. J., Joens, M. S., Fitzpatrick, J. A. J. Modern Laser Scanning Confocal Microscopy. *Current Protocols in Cytometry* **85**, 2018). We aim to present this work in more general terms as an exciting platform compatible with various mechanophore and matrix chemistries; therefore we elected to refer to images depicting mechanophore activation as fluorescence microscopy (FM) images.

Due to the addition of a few references to the manuscript, Ref. 33 is now Ref. 35. The publication year for Ref. 35 has been corrected.

REVIEWERS' COMMENTS

Reviewer #1 (Remarks to the Author):

The authors have adequately addressed all of my concerns. This study is interesting and important, and is now suitable for publication.

Reviewer #2 (Remarks to the Author):

The authors have responded to each of my previous comments and revised their manuscript. Although the detailed mechanism of mechanophore activation by shockwaves remains partially unresolved, as the authors have indicated, future studies involving various polymers and mechanophores, combined with simulations, will likely elucidate these mechanisms. Therefore, I recommend that this revised manuscript be published in Nature Communications.